# *Lactobacillus plantarum* HAC01 Supplementation Improves Glycemic Control in Prediabetic Subjects: A Randomized, Double-Blind, Placebo-Controlled Trial

**DOI:** 10.3390/nu13072337

**Published:** 2021-07-08

**Authors:** Mi-Ra Oh, Hui-Yeon Jang, Si-Yeon Lee, Su-Jin Jung, Soo-Wan Chae, Seung-Ok Lee, Byung-Hyun Park

**Affiliations:** 1Clinical Trial Center for Functional Foods, Chonbuk National University Hospital, Jeonju 54907, Korea; mroh@jbctc.org (M.-R.O.); janghy@jbctc.org (H.-Y.J.); sylee@jbctc.org (S.-Y.L.); sjjeong@jbctc.org (S.-J.J.); swchae@jbctc.org (S.-W.C.); 2Biomedical Research Institute, Chonbuk National University Hospital, Jeonju 54907, Korea; 3Division of Gastroenterology and Hepatology, Department of Internal Medicine, Chonbuk National University Medical School, Jeonju 54896, Korea; 4Department of Biochemistry and Molecular Biology, Chonbuk National University Medical School, Jeonju 54896, Korea

**Keywords:** *Lactobacillus plantarum*, HbA1c, clinical trial, 2h-PPG, prediabetes

## Abstract

A recent animal study demonstrated that administration of *Lactobacillus plantarum* HAC01 isolated from Korean kimchi improved glycemic control in type 2 diabetic mice. In the present study, we evaluated *Lactobacillus plantarum* HAC01’s effects on metabolic parameters of prediabetic human subjects. Forty subjects with isolated impaired glucose tolerance were randomly assigned to receive a daily placebo (*n* = 20) or a dose of *Lactobacillus plantarum* HAC01 (*n* = 20) over eight weeks. The primary endpoint was a change in 2 h postprandial glucose (2h-PPG) levels and the secondary endpoints were assessment of other glucose metabolism parameters, including HbA1c, gut microbiota composition, and fecal short-chain fatty acids (SCFAs). The group with a diet supplemented with *Lactobacillus plantarum* HAC01 saw a significant reduction in 2h-PPG and HbA1c levels compared to the placebo group. Fasting plasma glucose, insulin, HOMA-IR, QUICKI, microbiota composition, and fecal SCFAs, however, were not significantly altered. No serious adverse effects were reported. This is the first clinical trial to show a beneficial effect of single-strain probiotic supplementation administered over eight weeks on HbA1c levels in prediabetic subjects.

## 1. Introduction

Type 2 diabetes (T2D) has become much more prevalent in the last two decades, particularly in young people [1]. According to the International Diabetes Federation, 463 million people were diabetic in 2019, with this number expected to increase to 578 million by 2030 [2]. With proper management, patients can prevent cardiovascular sequelae, including coronary heart disease, stroke, renal failure, and retinopathy [3]. Type 2 diabetes treatments currently focus on alleviating insulin resistance through lifestyle changes and medication. It is difficult, however, to achieve adequate metabolic control with the current generation of treatments, and many patients ultimately fail to respond satisfactorily [4]. There is an urgent need for alternative strategies that both normalize blood glucose levels and increase the rate of treatment success.

Recent meta-analyses have confirmed that alterations to the composition of gut microbiota contribute to the development of insulin resistance in humans [5]. A diminished presence of *Firmicutes* and relatively higher levels of *Bacteroidetes* are found in type 2 diabetic patients, as compared to healthy individuals [6]. Altering the composition of gut microbiota through probiotics has attracted a great deal of attention among researchers seeking new means of controlling insulin resistance. *Bifidobacterium* and *Lactobacillus* species are the gut microbiota most commonly found in probiotics, and their ability to improve glucose homeostasis has been reported in several clinical trials [7,8,9,10]. Park et al. [11] recently isolated *Lactobacillus plantarum* HAC01 from Korean kimchi and observed its weight-lowering effect in high-fat diet-fed mice. The administration of *L. plantarum* HAC01 also improved glycemic control by restoring gut microbiota composition in diabetic mice, in which T2D was induced by a high-fat diet after the onset of pancreatic dysfunction by streptozotocin administration [12]. These reports indicated that this strain may have the potential to treat T2D in humans. To test this hypothesis, we conducted a randomized, double-blind, placebo-controlled trial in prediabetic subjects. Our primary aim was to investigate the effects of *L. plantarum* HAC01 on 2 h postprandial glucose (2h-PPG) levels. Our secondary aims were to investigate its effects on other glucose metabolism-related parameters, gut microbiota composition, and fecal short-chain fatty acids (SCFAs).

## 2. Materials and Methods

### 2.1. Ethics

This study adhered to Consolidated Standards of Reporting Trials (CONSORT) guidelines and was conducted according to the World Medical Association Helsinki Declaration (Version 2013). All subjects gave their written, informed consent before entering the study. The study protocol was approved by the Institutional Review Board of Chonbuk National University Hospital (CUH 2019-03-037) and was registered at the Clinical Research Information Service (CRIS) as No. KCT0005652.

### 2.2. Subjects

A total of 96 participants visited the site, 40 of whom met the inclusion criteria, defined as follows: (1) between 19 and 70 years old; (2) isolated impaired glucose tolerance (I-IGT), which is defined by a 2-h PPG level of between 140 and 199 mg after a 75 g oral glucose tolerance test (OGTT); and (3) voluntary agreement to participate in the clinical trials and provision of informed consent. Major exclusion criteria were: (1) history of underlying diabetes mellitus; (2) fasting plasma glucose (FPG) levels ≥ 140 mg/dL or HbA1c ≥ 7.0%; (3) administration of antibiotic within two weeks or corticosteroids in the four weeks prior to the screening visit; (4) administration of hypoglycemic agents, antiobesity drugs, and lipid-lowering agents, etc., in the three months prior to the screening visit; and (5) acute cardiovascular diseases, including myocardial infarction, cardiac insufficiency or stroke, liver failure, acute or chronic renal failure, history of alcohol or drug abuse, pregnancy, or lactation.

### 2.3. Study Design

This randomized, double-blind, placebo-controlled clinical trial was conducted between July 2019 and November 2020 in the Clinical Trial Center for Functional Foods (CTCF2) at Chonbuk National University Hospital. Subjects (*n* = 40) were randomly allocated into *L. plantarum* HAC01 or placebo groups by a computer-generated random sequence. The intervention was conducted over an eight-week period, during which subjects received either *L. plantarum* HAC01 or placebo capsules at 4-week intervals, with all subjects instructed to take one capsule per day after their main meal (breakfast, lunch, or dinner). All subjects recorded their daily intake of probiotics or fermented milk products, and subjects were discontinued if they consumed other probiotics or fermented milk products more than four times a week. Subjects were recommended to keep their routine lifestyle, including physical activity and dietary pattern during the eight-week intervention period.

Each test capsule contained a 4 × 10^9^ colony-forming unit (CFU) of *L. plantarum* HAC01. Dosing was based on the results of a previous animal study that showed improved glycemic control and accounted for the acceptable daily intake of *Lactobacillus* for humans without appreciable side effects [12]. The placebo capsule was composed primarily of microcrystalline cellulose; the flavor, color, appearance, and dosage of the two types of capsules were identical.

The study consisted of a screening visit and three additional visits (V1-V3, 42 days apart between each visit). Demographic data, medical history, and concomitant medications were recorded at the screening visit. OGTTs were also performed to confirm that eligibility criteria were satisfied. At Visit 1 (Day 0) and Visit 3 (Day 84), fecal and blood samples were collected. Vital signs, anthropometric data, physical activity, dietary intake, alcohol consumption, and adverse events were assessed at every visit. Fecal samples were obtained either a day prior to the visit or on the morning of the visit; in the former case, the subjects stored their fecal sample in a conventional freezer until the following morning. All subjects submitted approximately 5 g of fecal samples in a dedicated stool collection kit containing an ice pack. Fecal samples were stored at −70 °C and extracted by the addition of 1 mL of ice-cold 100% methanol. Each sample was homogenized (frequency = 30 Hz) for 10 min with a Retsch MM400 mixer mill (Retsch GmbH, Haan, Germany), after which each sample was centrifuged for 10 min at 4 °C and 17,000 rpm. The supernatants were passed through a 0.2 μm PTFE filter and transferred to Eppendorf tubes, then completely dried with a speed vacuum machine.

### 2.4. Biochemical Measurements

OGTTs were administered using a standardized protocol at the screening visit and Visit 3 (Day 84). Briefly, after an overnight fast, subjects consumed a 75 g glucose solution, and blood samples were collected from the median cubital vein and placed in sodium fluoride tubes at 0, 30, 60, 90, and 120 min after glucose loading. The 0 min blood sample was used to determine FPG, insulin levels, HbA1c, triglyceride (TG), total cholesterol (TC), HDL-cholesterol (HDL-C), LDL-cholesterol (LDL-C), adiponectin, and leptin. Glucose metabolism-related parameters were evaluated using the plasma glucose absolute maximum concentration (C_max_) and incremental area under the curve (iAUC). The iAUC was calculated using the trapezoidal rule for plasma glucose for each subject [13]. Plasma glucose levels were measured using the glucose oxidase method. Insulin concentration was measured using a Cobas e601 module (Hitachi High-Technologies, Tokyo, Japan). HbA1c levels were measured using an automated glycosylated hemoglobin analyzer (ARKRAY Factory, Shiga, Japan). Insulin resistance was estimated through a homeostasis model assessment [14]: HOMA-IR = [FPG (mg/dL) × fasting insulin (μU/mL)/405]. Insulin sensitivity was estimated by a quantitative insulin sensitivity check index (QUICKI) [15]: QUICKI = 1/[log(fasting insulin (μU/mL)) + log(FPG (mg/dL))]. Lipid profiles were measured with a Hitachi 7600-110^®^ analyzer (Hitachi High-Technologies). Adiponectin was measured by enzyme-linked immunosorbent assay (Molecular Devices, San Jose, CA, USA) and leptin levels were quantified by a double-antibody radioimmunoassay method (Molecular Devices).

### 2.5. Analysis of Fecal Microbiota

Fecal samples were obtained from all 40 subjects prior and subsequent to the intervention. Total fecal DNA was extracted using a QIAamp PowerFecal Pro DNA Kit (Qiagen, Hilden, Germany). The V4 hypervariable regions of the bacterial 16S rRNA were amplified with unique 8 bp barcodes and sequenced on the Illumina MiSeq PE300 platform [16]. Raw reads were analyzed using the QIIME pipeline [17]. Sequences were filtered and clustered into operational taxonomic units at 97% sequence identity according to the SILVA 132 database [18]. The operational taxonomic units were identified at the phylum to genus levels. The weighted UniFrac distances were used for PCoA [19].

### 2.6. Analysis of Fecal Short-Chain Fatty Acid

Feces samples (1.4 mg) were extracted by the addition of 25 μL of 3-nitrophenylhydrazine and 25 μL of N-3-dimethylaminopropyl-N′-ethylcarbodiimide. Each mixture sample was homogenized for 30 min with a Retsch MM400 mixer mill (Retsch GmbH). After centrifugation at 14,000 rpm for 5 min, LC-Triple Q-MS analysis (analytical concentration = 2.8 mg/mL) was performed on a Waters BEH C_18_ UPLC column (1.7 µm, 2.1 mm × 100 mm, Waters, Milford, MA, USA), using water/formic acid (100:0.01, *v/v*; solvent A) and acetonitrile/formic acid (100:0.01, *v/v*; solvent B) as the mobile phase for gradient elution.

### 2.7. Sample Size and Statistical Analysis

We determined the necessary sample size pursuant to methods developed in prior similar studies [20,21]. Based on a desired power of 80%, an alpha error probability of 0.05, and an estimated effect size of d = 1.3, we determined that 32 subjects were required. Anticipating a potential dropout rate of 20%, we enrolled 40 subjects in this study.

All analysis was performed based on the per-protocol (PP) set approach with the use of the SAS statistical software version 9.4 (SAS Institute, Cary, NC, USA). Categorical variables were summarized by frequency and proportions, and continuous variables were summarized by the mean and standard deviation (SD) or median and interquartile range (IQR). For between-group comparisons, the χ^2^ or Fisher exact test was used for categorical variables, and the independent samples *t*-test was used for differences between continuous variables as appropriate. A value of *p* < 0.05 was considered significant.

## 3. Results

### 3.1. Subjects Characteristics

Forty subjects were randomly assigned into two groups receiving either *L. plantarum* HAC01 (*n* = 20) or placebo (*n* = 20). After randomization, three subjects were excluded from the analysis set due to withdrawn consent (*n* = 2) and administration of a prohibited drug (*n* = 1). The per-protocol set, therefore, ultimately included a total of 37 subjects (Figure 1). Baseline characteristics were similar between the two groups (as shown in Table 1).

### 3.2. Parameters of Glucose Metabolism

Table 2 and Table 3 and Figure 2 present the change in parameters related to glucose metabolism, including FPG, 2h-PPG, iAUC_0–2h_, HbA1c, fasting insulin, HOMA-IR, and QUICKI. Compared to the placebo group, a significant reduction in 2h-PPG and HbA1c levels was observed in the group administered *L. plantarum* HAC01 after 8 weeks (Table 1 and Figure 2). There were no significant differences between the *L. plantarum* HAC01 and placebo groups in FPG, fasting insulin, HOMA-IR, and QUICKI (Table 2 and Table 3).

### 3.3. Lipid Profiles, Adiponectin, and Leptin

After eight weeks of intervention, there were no significant differences in TC, HDL-C, LDL-C, TG, adiponectin, and leptin between the *L. plantarum* HAC01 and placebo groups (data not shown).

### 3.4. Fecal Microbiota Composition and SCFAs

To determine the effects of *L. plantarum* HAC01 supplementation on the fecal microbial community, the bacterial composition in the feces was analyzed. Using a weighted UniFrac distance matrix, beta-diversity was examined through PCoA, with the results showing no significant difference between the study groups (data not shown). Consistent with these results, the fecal concentration of total SCFAs, as well as acetate, propionate, iso-butyrate, butyrate, iso-valerate, and valerate did not significantly differ between the study groups (data not shown).

### 3.5. Safety

Seven adverse events occurred during the intervention, none of which were serious. The proportion of subjects reporting an adverse event was similar in each study group (*L. plantarum* HAC01, *n* = 3; placebo, *n* = 4). Other safety parameters (vital signs, ECG, and laboratory data) in the *L. plantarum* HAC01 group were not significantly altered over the course of the study period (data not shown).

## 4. Discussion

The American Diabetes Association (ADA) defines prediabetes as existing when one of three distinct conditions are satisfied: impaired fasting glucose (IFG, FPG levels of 100–125 mg/dL), IGT (2h-PPG levels of 140–199 mg/dL during an OGTT), or HbA1c (5.7–6.4%) [22]. IFG and IGT can appear in isolation (I-IFG or I-IGT) or in combination (IFG + IGT). The ADA’s expert panel estimates that up to 70% of prediabetic individuals will eventually be diagnosed with T2D [23]. Thus, it is important to encourage prediabetic subjects to keep a healthy lifestyle to prevent or delay the onset of T2D. Both lifestyle and pharmacological interventions may improve this situation if implemented prior to the development of diabetes [24,25,26]. In this study, we specifically targeted patients with I-IGT, as this category of prediabetes is predominantly associated with unhealthy eating habits and physical inactivity, in contrast with I-IFG, which is more closely associated with male sex and other genetic factors [27,28]. Daily administration of *L. plantarum* HAC01 to subjects with I-IGT over an eight-week period significantly improved 2h-PPG (primary endpoint) and HbA1c (secondary endpoint) levels as compared to the placebo group. There was, however, no significant difference in FPG levels between the study groups, suggesting that *L. plantarum* HAC01 may be effective in prediabetic subjects with I-IGT, but not those with I-IFG.

Recent clinical trials exploring the use of various formulations of multistrain probiotics over 12-week periods resulted in significant reductions in HbA1c levels in subjects with T2D [10,29,30]. In contrast, two previous clinical trials using single-strain probiotics (*Lactobacillus casei* or *Lactobacillus reuteri*) failed to reduce HbA1c levels [9,31]. Taken as a whole, reports suggest that multistrain probiotics are more effective than single-strain probiotics at HbA1c reduction.

We observed a clear improvement in HbA1c levels after only eight weeks of *L. plantarum* HAC01 supplementation. As this finding was surprising, and aware that probiotics achieve their metabolic benefits through the modulation of the gut microbiota composition [32], we analyzed the gut microbiota composition. In contrast with the results of our animal study, in which *L. plantarum* HAC01 increased the presence of the *Akkermansiaceae* family and decreased the presence of the *Desulovibrionaceae* family in mice feces [12], we observed no changes in the abundance or taxonomic composition of the human fecal microbiome. These results suggest that it is microbial products, and not a difference in microbiota composition, that perform a regulatory role in glycemic control in humans. However, the interpretation of the results related to the microbial composition of the feces must proceed with caution, as fecal microbiota may not accurately reflect intestinal mucosa-associated microbiota, and prolonged storage of stool specimens, even at −70 °C, may have led to changes in the composition of the samples [33,34].

Recent studies have implicated SCFAs (including acetate, propionate, and butyrate) as primary microbial-derived metabolites that connect gut microbiota with their health-promoting effects [35]. SCFA binds to the receptors of free fatty acids, such as GPR41 and GPR43, and increases insulin sensitivity and pancreatic β-cell proliferation [36]. A comparison of baseline and Week 8 values confirmed no changes in the plasma concentrations of various SCFAs in subjects who received *L. plantarum* HAC01, suggesting that circulating SCFAs derived from gut microbiota were likely not responsible for the observed improvements in HbA1c levels. However, this conclusion warrants further investigation, as SCFAs have elsewhere been proposed as the most probable mechanism by which probiotics promote health outcomes [35] and fecal levels of SCFAs are affected by factors as diverse as microbiota composition, colonic transit time, luminal pH, host health status, and colonic clearance [37,38,39]. Moreover, the number of subjects in each group we sampled was small. Accordingly, our results should be treated as preliminary. Larger scale studies will validate these issues and assess the exact mechanism(s) by which *L. plantarum* HAC01 influences glucose parameters.

In this study, the primary endpoint was a change in 2h-PPG levels and intergroup comparison showed a significant improvement by *L. plantarum* HAC01 supplementation (*p* = 0.045). However, an intragroup comparison revealed no significant improvement in 2h-PPG levels after 8 weeks of *L. plantarum* HAC01 supplementation. Subjects in the placebo group exhibited relatively low baseline levels of 2h-PPG compared with the *L. plantarum* HAC01-supplemented group. These results indicate that the observed improvement in 2h-PPG levels was not a result of *L. plantarum* HAC01’s insulin sensitizing effects. Seemingly confirming this, we observed that plasma insulin levels, HOMA-IR, and QUICKI were also unchanged by the administration of *L. plantarum* HAC01.

This study has some limitations. Though sufficiently powered, the total number of subjects that participated was not large. As the primary aim of this study was to determine whether single-strain probiotic *L. plantarum* HAC01 was beneficial to prediabetic subjects, the study size was sufficient to achieve this goal. Additionally, we did not measure a dose-response relationship. Further studies with a larger sample size and multiple doses will be required to determine whether *L. plantarum* HAC01 is capable of consistently achieving adequate glycemic control in prediabetic humans.

## 5. Conclusions

This clinical study confirmed the results of recent animal studies and demonstrated that an eight-week course of *L. plantarum* HAC01 supplementation significantly improved HbA1c and 2h-PPG levels relative to placebo in prediabetic subjects. No serious adverse effects were observed, suggesting that *L. plantarum* HAC01 has potential as an effective lifestyle intervention to forestall or prevent the onset of T2D.

## Figures and Tables

**Figure 1 nutrients-13-02337-f001:**
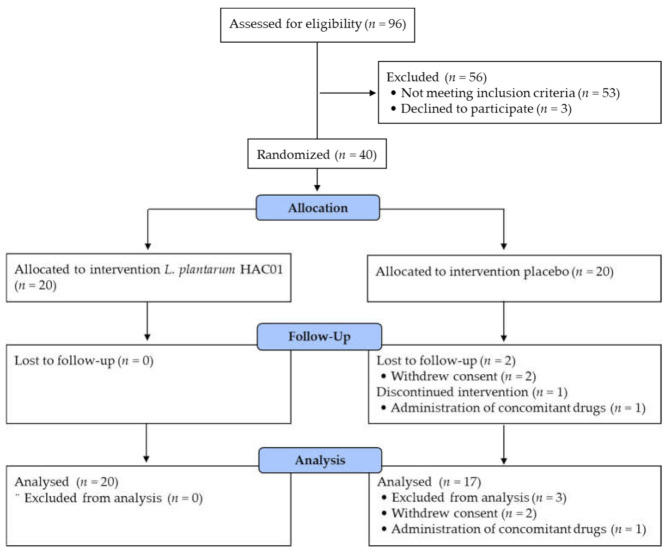
Flow diagram of subject enrollment, allocation, follow-up, and analysis.

**Figure 2 nutrients-13-02337-f002:**
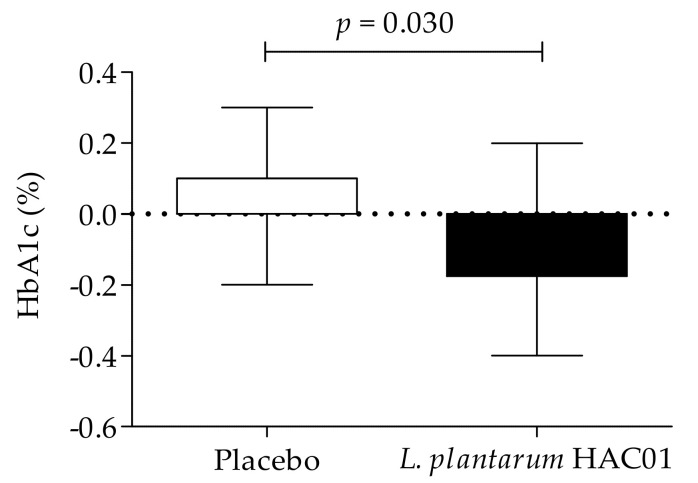
Effect of *L. plantarum* HAC01 supplementation on the change (%) of HbA1c levels from the baseline value (adjusted to zero) to Week 8.

**Table 1 nutrients-13-02337-t001:** Baseline participant clinical characteristics and demographics.

	Placebo (*n* = 20)	*L. plantarum* HAC01 (*n* = 20)	*p*-Value
Age, years	53.55 ± 10.18	56.40 ± 11.57	0.413
Sex, female, *n* (%)	17 (85.0)	14 (70.0)	0.451
Height, cm	159.90 ± 6.07	161.60 ± 8.74	0.479
Weight, kg	63.99 ± 5.68	66.53 ± 14.09	0.462
BMI, kg/m^2^	25.03 ± 1.92	25.25 ± 3.14	0.793
WHR			
Female	0.91 ± 0.05	0.93 ± 0.04	0.149
Male	0.91 ± 0.03	0.95 ± 0.02	0.066
Current smoker, *n* (%)	2 (10.0)	0 (0.0)	0.487
Alcohol consumption, unit/day	6.81 ± 5.16	7.32 ± 12.02	0.922
Physical activity, MET-min/week	1940 (830–2720)	980 (440–3060)	0.762
FPG, mg/dL	101.65 ± 8.21	99.00 ± 5.90	0.249
2h-PPG, mg/dL	161.95 ± 14.69	172.15 ± 19.09	0.066
HbA1c, %	5.94 ± 0.38	5.93 ± 0.33	0.930

BMI, body mass index; WHR, waist–hip circumference ratio; MET, metabolic equivalent of task; FPG, fasting plasma glucose; PPG, postprandial plasma glucose. MET is presented as the median (interquartile range).

**Table 2 nutrients-13-02337-t002:** Change in blood glucose levels during the intervention period.

		Placebo (*n* = 17)	*L. plantarum* HAC01 (*n* = 20)	*p*-Value ^(1)^
FPG, mg/dL	Baseline	101.65 ± 8.44	99.00 ± 5.90	0.271
	8 weeks	101.47 ± 10.36	97.45 ± 8.48	0.625
	Change value	−0.18 ± 10.09	−1.55 ± 5.89	
	*p*-value ^(2)^	0.943	0.254	
0.5h-PPG, mg/dL	Baseline	172.71 ± 32.64	166.15 ± 22.77	0.478
	8 weeks	166.24 ± 25.36	160.25 ± 28.80	0.946
	Change value	−6.47 ± 27.22	−5.90 ± 23.69	
	*p*-value ^(2)^	0.342	0.279	
1h-PPG, mg/dL	Baseline	194.18 ± 37.84	191.10 ± 25.91	0.772
	8 weeks	192.88 ± 23.51	187.10 ± 29.67	0.795
	Change value	−1.29 ± 35.42	−4.00 ± 27.28	
	*p*-value ^(2)^	0.882	0.520	
1.5h-PPG, mg/dL	Baseline	181.12 ± 33.82	182.80 ± 27.59	0.869
	8 weeks	188.88 ± 32.46	185.00 ± 28.84	0.620
	Change value	7.76 ± 38.21	2.20 ± 29.48	
	*p*-value ^(2)^	0.414	0.742	
2h-PPG, mg/dL	Baseline	163.12 ± 14.36	172.15 ± 19.09	0.118
	8 weeks	182.24 ± 30.08	170.30 ± 30.28	0.045 *
	Change value	19.12 ± 35.06	−1.85 ± 26.30	
	*p*-value ^(2)^	0.039 *	0.757	
iAUC_0–2h_, h·mg/dL	Baseline	136.90 ± 40.97	139.81 ± 32.65	0.811
	8 weeks	141.99 ± 33.18	138.28 ± 40.47	0.621
	Change value	5.09 ± 43.32	−1.53 ± 37.36	
	*p*-value ^(2)^	0.635	0.857	
HbA1c, %	Baseline	5.94 ± 0.38	5.93 ± 0.33	0.931
	8 weeks	5.98 ± 0.37	5.84 ± 0.33	0.013 *
	Change value	0.04 ± 0.14	−0.09 ± 0.15	
	*p*-value ^(2)^	0.248	0.020 *	

FPG, fasting plasma glucose; PPG, postprandial plasma glucose; iAUC, incremental area under the curve. *p*-value < 0.05 are denoted with an asterisk (*). ^(1)^ Compared between groups; *p*-value by independent *t*-test. ^(2)^ Compared within group; *p*-value by paired *t*-test.

**Table 3 nutrients-13-02337-t003:** Change of insulin, HOMA-IR, and QUICKI during the intervention period.

		Placebo (*n* = 17)	*L. plantarum* HAC01 (*n* = 20)	*p*-Value ^(1)^
Insulin, μU/mL	Baseline	11.54 ± 5.29	9.63 ± 4.24	0.230
	8 weeks	11.44 ± 6.17	9.53 ± 4.05	0.998
	Change value	−0.10 ± 5.34	−0.09 ± 3.06	
	*p*-value ^(2)^	0.941	0.892	
HOMA-IR	Baseline	2.93 ± 1.42	2.36 ± 1.06	0.166
	8 weeks	2.91 ± 1.60	2.35 ± 1.10	0.958
	Change value	−0.02 ± 1.56	0.00 ± 0.86	
	*p*-value ^(2)^	0.949	0.992	
QUICKI	Baseline	0.33 ± 0.03	0.34 ± 0.03	0.268
	8 weeks	0.34 ± 0.04	0.34 ± 0.03	0.761
	Change value	0.01 ± 0.03	0.00 ± 0.02	
	*p*-value ^(2)^	0.515	0.621	

HOMA-IR, homeostatic model assessment for insulin resistance; QUICKI, quantitative insulin sensitivity check index. ^(1)^ Compared between groups; *p*-value by independent *t*-test. ^(2)^ Compared within group; *p*-value by paired *t*-test.

## Data Availability

The datasets generated during and/or analyzed during the current study are available from the corresponding author on reasonable request.

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
