# Peer review of "Lactobacillus plantarum HAC01 Supplementation Improves Glycemic Control in Prediabetic Subjects: A Randomized, Double-Blind, Placebo-Controlled Trial"

_nutrients, 2021, doi:10.3390/nu13072337_

Round 1

Reviewer 1 Report

The findings are quite limited to support firm conclusions.

The authors reported that L. plantarum HAC01 capsules contained 15.94 mg per capsule, but it is not clear what this means. No information are available about bacterial growth conditions, detailed protocols for culturing, harvesting, freeze-drying, storing, and viable count determining of Lactobacillus, number of colony forming units.

How did the authors guarantee the similar appearance and taste for the placebo? How was a possible effective dosage established?

In Figure 1 placebo subject enrollment at follow-up is not clear.

In table 2 the authors should report the absolute values of HbA1c.

The primary endpoint was a change in 2 h postprandial glucose (2h-PPG) levels, but no improvement was observed in the group treated with L. plantarum HAC01. Rather, a worsening was observed in the placebo group. Therefore the primary endpoint was not reached.

It is not clear why some subjects in the placebo group developed diabetes in two months as evidenced by the OGTT, thus slightly modifying the HbA1c values. The only explanation seems to be that there is an increased insulin sensitivity which occurred only at the end of the second hour. How do the authors explain this?

How many cases of diabetes occurred in the two groups respectively after 8 weeks?

Figure 3 does not appear necessary

How were the authors sure that the patients did not change their diet in the two months? Have they collected any information in this regard?

Reviewer 2 Report

In this work, the authors determined the effects of Lactobacillus plantarum HAC01 on glucose homeostasis in patients with glucose intolerance. In the randomized double-blind placebo-controlled trial, they report that the single-strain probiotic HAC01 supplementation significantly reduced 2 h postprandial glucose (2h-PPG) and HbA1c levels, which are indicative of improved glucose homeostasis in these patients.

Major comments:

  • How was the sample number determined? Please indicate the methodology and whether the sample number is sufficiently powered.
  • Please explain that the 2h-PPG of the placebo group (182) is significantly higher than basal (163). In another word, if 2h-PPG of placebo group is similar to the basal 2h-PPG (163), the 2h-PPG (172) of the HAC01-treated group may not be different from the placebo group. Please explain.
  • Please include data on the effects of HAC01 on probiotic strains and short-chain fatty acids. It is possible that the sample number isn’t adequately powered to detect difference; however, the data would indicate trends of the changes, if there is any, which would be interesting to readers.

Round 2

Reviewer 1 Report

The authors have satisfactorily responded to all the questions.

Author Response

No specific response to reviewers